# Adsorption of Fatty Acid on Beta-Cyclodextrin Functionalized Cellulose Nanofiber

Nor Hasmaliana Abdul Manas [1,2,*], Nurhidayah Kumar Muhammad Firdaus Kumar [1],
Nurul Aqilah Mohd Shah [1], Guang Yik Ling [1], Nur Izyan Wan Azelee [1,2], Siti Fatimah Zaharah Mohd Fuzi [3],
Nasratun Masngut [4], Muhammad Abd Hadi Bunyamin [5], Rosli Md. Illias [1,2] and Hesham Ali El Enshasy [1,2,6]

1   Faculty of Chemical and Energy Engineering, Universiti Teknologi Malaysia, Skudai 81310, Johor, Malaysia
2   Institute of Bioproduct Development, Universiti Teknologi Malaysia, Skudai 81310, Johor, Malaysia
3   Department of Technology and Natural Resources, Faculty of Applied Sciences and Technology,
    Universiti Tun Hussein Onn Malaysia, Pagoh Campus, Panchor 84600, Johor, Malaysia
4   Faculty of Chemical and Process Engineering Technology, Universiti Malaysia Pahang, Lebuhraya Tun Khalil
    Yaakob, Kuantan 26300, Pahang, Malaysia
5   Faculty of Social Sciences and Humanities, Universiti Teknologi Malaysia, Skudai 81310, Johor, Malaysia
6   City of Scientific Research and Technology Applications (SRTA), New Burg Al Arab, Alexandria 21934, Egypt
*   Correspondence: hasmaliana@utm.my; Tel.: +60-7553-5691

**Abstract:** Fatty acids in wastewater contribute to high chemical oxygen demand. The use of cellulose nanofiber (CNF) to adsorb the fatty acids is limited by its strong internal hydrogen bonding. This study aims to functionalize CNF with β-cyclodextrin (β-CD) and elucidate the adsorption behaviour which is yet to be explored. β-CD functionalized CNF (CNF/β-CD) was achieved by crosslinking of β-CD and citric acid. Functionalization using 7% (*w/v*) β-CD and 8% (*w/v*) citric acid enhanced mechanical properties by increasing its thermal decomposition. CNF/β-CD was more efficient in removing palmitic acid, showcased by double adsorption capacity of CNF/β-CD (33.14% removal) compared to CNF (15.62% removal). CNF/β-CD maintained its adsorption performance after five cycles compared to CNF, which reduced significantly after two cycles. At 25 °C, the adsorption reached equilibrium after 60 min, following a pseudo-second-order kinetic model. The intraparticle diffusion model suggested chemical adsorption and intraparticle interaction as the controlling steps in the adsorption process. The maximum adsorption capacity was 8349.23 mg g$^{-1}$ and 10485.38 mg g$^{-1}$ according to the Sips and Langmuir isotherm model, respectively. The adsorption was described as monolayer and endothermic, and it involved both a physisorption and chemisorption process. This is the first study to describe the adsorption behaviour of palmitic acid onto CNF/β-CD.

**Keywords:** electrospinning; adsorption kinetics; adsorption isotherms; β-cyclodextrin; oily wastewater

## 1. Introduction

Oily wastewater is not a new issue when it comes to water pollution. A large amount of oily wastewater effluents was discharged into rivers by industries such as oil and gas, textile, food, and petrochemical processing. The discharge of oily effluent into the sea or river will have a negative influence on the ecosystem and living beings. A presence of oil in emulsified form in the oily wastewater is extremely challenging to remove from the aqueous media [1]. A droplet size smaller than 20 μm is a stable emulsion in the emulsified oily wastewater. The effective separation of the emulsified oily wastewater is one of the most challenging issues in the field of wastewater treatment because of its stable dynamic structure and nonuniform small droplet size [2]. The difficulty of treating the oily wastewater lies in the existence of the stable oil-in-water emulsions. The traditional treatment methods were rendered by the presence of the stable oil-in-water emulsions [3]. Nanofibrous materials hold excellent potential for various environmental applications, including in the treatment of wastewater, due to its high porosity, bigger surface area, and better connectivity [4]. Various types

of materials can be used, like polymer, ceramic and carbon, thus making the scalable synthesis of it much easier. Cellulose is inexpensive and comes as the most abundant natural bio polymer to be considered in developing a cost-effective wastewater treatment technology [5]. In addition to the expansion of nanotechnology for fabricating polymer-based materials, cellulose nanofiber (CNF) produced by an electrospinning technique seems to be a promising method for use in wastewater treatment. The key success of nanofibrous materials was due to its voids among fibers, which led to a better selectivity. Nanofibrous materials were said to have higher sorption capacity compared to non-nanofibrous or non-porous materials. However, the high hydrophilicity of this material [6] has become one of the main drawbacks; therefore, its surface needs to be modified in order to expand the application of CNF.

Cyclodextrins (CD) are a group of cyclic oligomers composed of α- (1,4) linked glucopyranose subunits having a 3D structure that looks like a cup or a shallow, truncated cone with a hydrophobic core and hydrophilic exterior. The capability of the CDs to act as hosts for guests in forming noncovalent host-guest inclusion complexes has made them an attractive compound [7]. α- cyclodextrin (α- CD), β- cyclodextrin (β-CD) and γ- cyclodextrin (γ- CD) are the most common native CDs and are differentiated by the number of glucose units. On the other hand, β-CD is a type of CD that has many advantages as it is cheap and accessible, and is thus the most studied CD [8]. Inclusion complexes, or host-guest complexes between CDs and other molecules or pollutants, are stabilized by weak forces, which implies an equilibrium between free and complex species [9]. However, CDs are soluble in water; therefore, it cannot be directly applied in water for adsorption of pollutants. In order to improve its performance, several works had been done on the immobilization of CDs on a good, insoluble support including nanopolymers. Surface functionalization of nanofibers with CD would be interesting for designing an efficient filtering material. β-CD functionalized nanofibers produced via electrospinning has drawn a positive result in improving the adsorption and separation of various types of pollutants from aqueous solution including dyes, polycyclic aromatic hydrocarbon (PAH), heavy metals, oil and others [10]. However, to the best of our knowledge, the study on β-CD functionalized CNF for removal of fatty acids has not been previously performed. Palmitic is one of the long chain fatty acids (LCFAs) found in high concentration in the palm oil mill effluent that is recalcitrant to biodegradation [11]. It was also a major fatty acid found in industrial dairy wastewater (65%) [12]. LCFAs at concentrations higher than 0.5 mM could potentially inhibit in anaerobic digester of wastewater treatment [13] and at concentrations above 16 mM was shown to cause a lag in methane production [14]. Therefore, there is an urgency in removing LCFAs from the wastewater.

In this study, the surface modification of CNF with β-CD (CNF/β-CD) was achieved by a polymerization reaction between β-CD and citric acid as crosslinking agent. Different concentrations of β-CD and citric acid were used in order to optimize the factors affecting functionalization process. The morphological, surface and thermal decomposition properties of CNF/β-CD were characterized. The adsorption performance of the CNF/β-CD was investigated by removal of a model fatty acid (palmitic acid) from aqueous solution. The adsorption kinetics and isotherms were elucidated to precisely describe the adsorption behaviour.

## 2. Materials and Methods

The materials and methodology for the study are described in the following subsection. All chemicals and materials were used without further purifcation.

### 2.1. Materials

Cellulose acetate powder (average $M_n$ ~30,000 by GPC), β-cyclodextrin (≥97% Sigma-Aldrich, Missouri, United States), citric acid, sodium hydrophosphite hydrate (SHPI), acetone, dimethylacetamide (DMAc), palmitic acid powder, 1-propanol (99% of purity).

### 2.2. Synthesis of CNF by Electrospinning Process

First, cellulose acetate powder was used as the polymer to form nanofibers that was then functionalized with β-CD. The optimum parameters used were based on the method adapted from Liu and Hsieh [15]. Cellulose acetate solution with concentration of 15% (*w/v*) was prepared in acetone: dimethylacetamide mixtures (2:1). Next, the electrospinning was carried out in which 8 mL of cellulose acetate solution was placed into a syringe fitted with a metallic needle having an inner diameter of 0.8 mm, and the syringe was horizontally placed at the syringe pump. The flow rate of polymer solution during electrospinning process was set at 1 mL h$^{-1}$ with 15 cm distance between needle tip and metal collector covered with a piece of aluminium foil. The voltage was set at 15 kV throughout the process, and it was carried out at room temperature in an enclosed Plexiglas box.

### 2.3. Functionalization of Cellulose Nanofiber with β-cyclodextrin

For functionalization of CNF with β-CD, different concentration of β-CD solutions (5%, 6%, 7%, 8%, 9% and 10% *w/v*) were prepared in 50 mL aqueous solution at 50 °C. Then, different concentrations of citric acid (5%, 6%, 7%, 8%, 9% and 10% *w/v*) and 1.25% *w/v* of sodium hydrophosphite hydrate (SHPI) that act as a catalyst were added to each β-CD solutions separately and stirred continuously using a magnetic stirrer with mixing speed of 150 rpm for about 30 min at 50 °C. After all reactants were completely dissolved in aqueous solution, a rectangular shaped 4 cm × 4 cm of CNF (weighed about 0.05 g) was immersed into each different mixture of β-CD solutions and kept at 50 °C for a duration of 3 h. Then, the β-CD functionalized cellulose nanofibers (CNF/β-CD) were dried at 70 °C for 3 h in order to ensure the crosslinking of β-CD with cellulose nanofibers. The dried CNF/β-CD mats were washed two times with 40 °C warm water for removal of unreacted β-CD and citric acid. Lastly, all of the mats were dried again at 70 °C for 3 h or until they achieved constant weight.

### 2.4. Morphological, Surface and Thermal Characterization of Nanofibers

Scanning electron microscope (SEM) was performed by using Hitachi TM3000 SEM to check the changes that occurred on the morphology and fiber diameter of CNF and CNF/β-CD mats. The CNF and CNF/β-CD mats were coated with 5 nm Au/Pd prior to SEM analysis, and each of the samples was captured at magnifications (×10,000) randomly at any spot to examine the average fiber diameter (AFD) of the nanofibers. It was very vital to ensure that the modification performed during crosslinking process did not deform the original structure of cellulose nanofibers.

Next, the surface chemical characterization was carried out by using Fourier transform infrared spectroscopy (FTIR) for both CNF and CNF/β-CD. Dried CNF and CNF/β-CD samples were pressed into thin transparent films. FTIR analysis was carried out by using a Shimadzu IRTracer- 100 (ATC) (Shimadzu Scientific Instruments, Maryland, United States) to observe changes of the functional groups on the surface of CNF/β-CD. Each spectrum was obtained with a wavelength in the range of 400 cm$^{-1}$ and 4000 cm$^{-1}$.

The thermogravimetric analyzer (TGA) was used to investigate the thermal characterization. CNF and CNF/β-CD were analysed by using Shimadzu TGA-50 equipment under a itrogen (N$_2$) atmosphere with a purge rate of 100 mL min$^{-1}$ with temperatures ranging from 50 °C to 900 °C at a heating rate of 10 °C min$^{-1}$. Samples of 6–8 mg were used for each test. This TGA analysis was applied for CNF and CNF/β-CD in order to study the effect of functionalization of CNF with β-CD on the thermal stability.

### 2.5. Adsorption of Palmitic Acid

2.5.1. Comparison of Palmitic Acid Adsorption onto CNF and CNF/β-CD

The performance of the CNF and CNF/β-CD mats as adsorbent for palmitic acid was tested. Identical square-shaped CNF and CNF/β-CD mats weighing 0.1 g were immersed individually into 30 mL of palmitic acid solution with a concentration of 70,000 ppm in order to study the adsorption performance. The reduction of palmitic acid concentration

was checked at different contact times (15 min, 30 min, 60 min, 90 min and 120 min), and the percentage of removal was calculated.

Reusability of the CNF and CNF/β-CD were also investigated. After immersing the CNFs in the palmitic acid solution for the first time, it was then washed with 50 mL 1-propanol at 50 °C for 3 h in a thermostatic water bath shaking incubator and dried in a vacuum oven at 70 °C for 3 h or until it was completely dried. These washing steps were repeated for two times in order to ensure the previous palmitic acid captured was totally detached from the CNF and CNF/β-CD. Next, the CNFs were immersed again in the 30 mL, 70,000 ppm palmitic acid solution, and the reduction of the palmitic acid concentration was monitored for 2 h. The changes on the adsorption performance for each cycle was recorded. The reusability study was continuously carried out until both CNFs show a significant drop in adsorption performance.

### 2.5.2. Adsorption Kinetics

The 5% (*w/v*) palmitic acid solution was prepared by dissolving 5 g of palmitic acid powder into 100 mL 1-propanol (99% of purity). The mixture was then continuously stirred by using a magnetic stirrer with 150 rpm at room temperature for about 30 min to have a well-mixed solution. Identical square-shaped CNF and CNF/β-CD mats weighing 0.1 g were immersed individually into 30 mL of palmitic acid solution with a concentration of 70,000 ppm at room temperature (25 °C). The reduction of palmitic acid concentration was checked at different contact time (15 min, 30 min, 60 min, 90 min and 120 min). The percentage removal efficiency and adsorption capacity ($q_e$) of pollutant by the adsorbent were calculated as described in Equation (1) and (2).

$$\text{Removal efficiency (\%)} = \frac{C_0 - C_t}{C_0} \times 100 \tag{1}$$

$$q_e \text{ (mg/g)} = \frac{(C_0 - C_t) \times V}{W} \tag{2}$$

where $C_0$ (mg L$^{-1}$) is the concentration of pollutants at initial, $C_t$ (mg L$^{-1}$) is the concentration of pollutants at certain time, W (g) is the weight of the adsorbent and V (L) is volume of the testing solution.

The kinetic behavior of the adsorption process was investigated by nonlinear data fitting into the pseudo-first-order model [16], pseudo-second-order model [17] and Elovich model [18] as in Equation (3)–(5) respectively.

$$q_t = q_e \left(1 - e^{-k_1 t}\right) \tag{3}$$

$$q_t = \frac{t}{\frac{t}{q_e} + \frac{1}{k_2 q_e^2}} \tag{4}$$

$$q_t = \frac{1}{\beta \ \ln(\alpha \ \beta \ t + 1)} \tag{5}$$

where $q_t$ and $q_e$ (mg g$^{-1}$) represent the adsorption capacity at certain time and equilibrium time respectively, while $k_1$ (min$^{-1}$) and $k_2$ (g mg$^{-1}$ min$^{-1}$) are the pseudo-first-order model rate constant and pseudo-second-order model rate constant respectively. For the Elovich model, $\alpha$ is the initial adsorption rate (mg g$^{-1}$ min$^{-1}$), and $\beta$ is a desorption constant. Adsorption diffusion was investigated by using Weber–Morris intraparticle diffusion model [19] as shown in Equation (6).

$$q_t = k_{id} t^{1/2} + C \tag{6}$$

where $k_{id}$ is the intraparticle diffusion rate constant and $C$ is the boundary effect.

2.5.3. Adsorption Isotherms and Thermodynamic

CNF/β-CD weighing 0.1 g was immersed in various concentrations of palmitic acid solutions (10,000, 30,000, 50,000, 70,000 and 90,000 mg L$^{-1}$)for 60 min. The final concentration of palmitic acid in the solutions was measured. Adsorption isotherm was investigated using non-linear data fitting into Langmuir [20], Freundlich [21], Sips [22] and Temkin [23] isotherm models, where the equation for the models are shown in Equations (7)–(10) respectively.

$$q_e = q_{max} \frac{K_L C_e}{1 + K_L C_e} \tag{7}$$

$$q_e = K_F C_e^{1/n_F} \tag{8}$$

$$q_e = \frac{q_m (K_s C_e)^{n_s}}{1 + (K_s C_e)^{n_s}} \tag{9}$$

$$q_e = b_T \ln(A K_T C_e) \tag{10}$$

where $q_e$ (mg g$^{-1}$) is the amount of fatty acids adsorbed, $q_m$ (mg g$^{-1}$) is the maximum adsorption, $K_L$ (L mg$^{-1}$) is the Langmuir coefficient, $C_e$ (mg L$^{-1}$) is the equilibrium concentration of fatty acids, $K_F$ and $n_F$ are the Freundlich constants, $K_s$ (mg$^{-1}$) and $n_s$ is the Sips equilibrium constants, $b_T$ (J mol$^{-1}$) is heat of adsorption and $K_T$ (L mg$^{-1}$) is the equilibrium binding constant. Dimensionless constant separation factor, $R_L$ for Langmuir isotherm model was calculated using Equation (11) [24].

$$R_L = \frac{1}{1 + K_L C_0} \tag{11}$$

where $K_L$ (L mg$^{-1}$) is the Langmuir coefficient and $C_0$ (mg L$^{-1}$) is the initial concentration. The Gibbs free energy of the adsorption was calculated using Equation (12) [25] to measure the spontaneity of the process.

$$\Delta G^o = -RT \ln K_L \tag{12}$$

where R is the gas constant (8.314 J mol$^{-1}$ K$^{-1}$), T is the absolute temperature (K) and $K_L$ is the Langmuir equilibrium constant (L mg$^{-1}$).

*2.6. Palmitic Acid Quantification Using High Performance Liquid Chromatography*

The concentration of palmitic acid in the solution during the time course of the adsorption experiment was measured by using Agilent 1200 Infinity Series high performance liquid chromatography (HPLC) with C18 column. Acetonitrile and isopropanol (80:20 *v/v*) at a flow rate of 1 mL min$^{-1}$ were used as the mobile phase, and the concentration of palmitic acid taken at different contact times was monitored by the UV detector at the wavelength of 210 nm [17]. Each of the samples of palmitic acid solution taken at different contact times was filtered by using 0.45 μm disposable syringe filter and injected into 1.5 mL autosampler HPLC vials. As a result, the amount of remaining palmitic acid in each sample was measured from the area of palmitic acid peak observed in HPLC chromatograms. Palmitic acid solutions with different concentrations ranging from 0.5% to 5% (*w/v*) were prepared to obtain the calibration curve and $R^2$ was calculated as 0.9995.

## 3. Results and Discussion

*3.1. The Synthesis of Beta-Cyclodextrin Functionalized Cellulose Nanofiber*

The optimum parameters used to synthesize cellulose nanofiber (CNF) via electrospinning process in this study was found to be sufficient to form smooth and continuous CNF with an average weight of 0.05 g for each 4 cm × 4 cm dimension of the CNF mats. Acetone and dimethylacetamide (DMAc) with a ratio of 2:1 were used as solvents for producing 15% (*w/v*) cellulose acetate polymer. Acetone and DMAc that have solubility of (9.76 cal cm$^{-3}$)$^{1/2}$ and 11.1 (cal cm$^{-3}$)$^{1/2}$, respectively, were used as solvents, as they fit the solubility required to be solvent for cellulose acetate where the Hildebrand solubility

parameter ($\delta$) should be in a range between 9.5 and 12.5 (cal cm$^{-3}$)$^{1/2}$ [15]. Moreover, the low surface tension offered by acetone (23.7 dyne cm$^{-1}$) and DMAc (32.4 dyne cm$^{-1}$) generated a mixture of these solvents that is suitable and efficient for electrospinning. Cellulose acetate with concentration range between 12.5 and 20% in 2:1 acetone: DMAc could form a good fibrous membrane due to high viscosity of the polymer liquids [15]. The jet produced by high viscosity liquid does not break up but travels as a jet to the grounded target and thus forms continuous fibers.

### 3.2. Functionalization of Cellulose Nanofiber with β-Cyclodextrin

Surface modification of the resulted CNF from electrospinning was achieved through crosslinking reaction between β-CD and citric acid, and no leaching of β-CD was found when the washing process was carried out. Surface modification of CNF is very essential in order to synthesize a better nanomaterial adsorbent. Different concentrations of β-CD and citric acid as crosslinking agents were the parameters studied in this case to synthesize the best adsorbent for removal of palmitic acid that acts as the model of fatty acids pollutants. The impregnation of CNF with different concentrations of β-CD (5, 6, 7, 8, 9 and 10% (*w/v*)) (Figure 1a) while the concentration of citric acid was maintained at 5% (*w/v*) during the crosslinking process, influenced the texture, structure and performance of the CNF that were used as adsorbent for palmitic acid removal. The amount of palmitic acid being removed increases with increasing concentration of β-CD. This is because larger amount of β-CD attached on the surface of CNF after the functionalization process will eventually lead to more binding between β-CD and palmitic acid due to more vacant adsorption sites [26]. The percentage rate of removal of palmitic acid for CNF and CNF/β-CD functionalized with 7% (*w/v*) of β-CD concentration were 0.73% min$^{-1}$ and 1.87% min$^{-1}$, respectively.

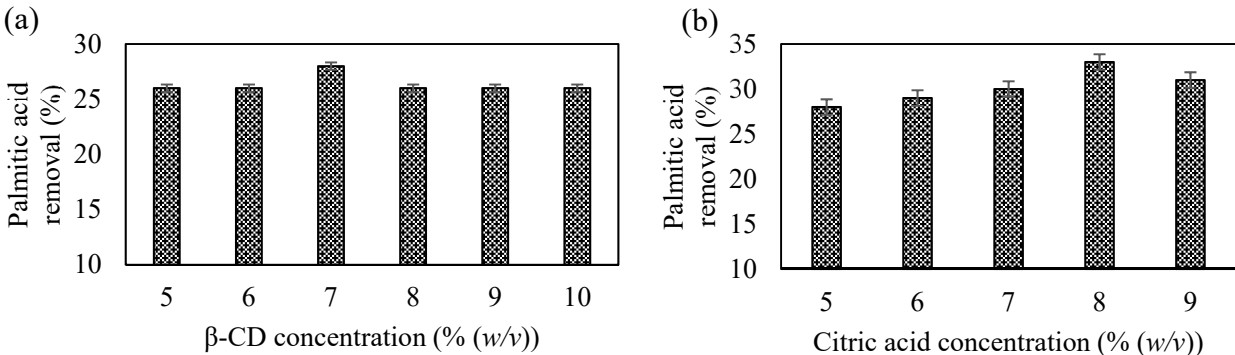

**Figure 1.** The effect of (**a**) β-CD concentration and (**b**) citric acid concentration on palmitic acid removal. The error bars represent the standard deviation of triplicate experiments.

Next, an increase in concentration of citric acid (5, 6, 7, 8, 9 and 10% (*w/v*)) (Figure 1b) during the crosslinking process was assumed to increase the carboxyl content of the CNF. Therefore, with a higher concentration of citric acid introduced during the crosslinking process, more β-CD will be attached to the CNF which acts as molecules that captured the palmitic acid and hence increase the reduction of palmitic acid concentration from the aqueous solution. It was found that 8% (*w/v*) was the optimum citric acid concentration to crosslink between CNF and 7% (*w/v*) of β-CD. The highest removal percentage of palmitic acid concentrations performed by CNF and optimum CNF/β-CD were 16% and 33%, respectively. Moreover, the concentration of palmitic acid became constant or achieved equilibrium after 60 min of contact time. Lastly, it could be concluded that a 7% of β-CD functionalized CNF with a dimension size of 4 cm × 4 cm needs 8% concentration of citric acid as a crosslinker to achieve the highest removal efficiency of palmitic acid at 60 min contact time. It clearly shows that the removal of palmitic acid amount from its aqueous solution was better when CNF/β-CD mats were used.

### 3.3. Morphological Characterization of the Nanofibers

Scanning electron microscope (SEM) analysis was performed to investigate the changes of the morphology of the CNF after the modification was made on its surface with β-CD during the functionalization process. The images of the morphology and its fiber diameter are represented as in Figure 2. The surface morphology for CNF and CNF/β-CD were obviously different, as clearly seen from the SEM images. For the CNF, it has a smooth and uniform surface structure compared to the CNF/β-CD that appeared rough due to the functionalized β-CD. Irregularities at certain point of CNF/β-CD were also observed. Resulting rough surface and irregularities of the nanofibers were also reported for modified electrospun polyester with CDs [27], cotton fabrics grafted with glycidyl methacrylate/β-CD [28] and woven PET vascular prosthesis grafted with CD [29]. Therefore, the roughness and irregularities on the surface of CNF indicated the successful attachment of β-CD onto CNF.

It is vital to ensure the functionalization process made on CNF did not deform the fibrous structure of CNF. This study has proven that functionalized CNF with β-CD maintained the fibrous structure as clearly seen from the SEM images. In addition, CNF was recorded to have a range of the average fiber diameter from 133 nm to 241 nm, and this result was supported by a finding from the study of preparation of cellulose- based nanofibers using electrospinning [30], where the fiber diameter generated from 15kV of electrospinning condition was between 100 nm to 200 nm. On the other hand, CNF/β-CD was observed to have larger average fiber diameter ranges from 262 nm to 378 nm. The increases of fiber diameter for after modification made on CNF was also demonstrated by a study for modified electrospun polyester with cyclodextrin polymer [27].

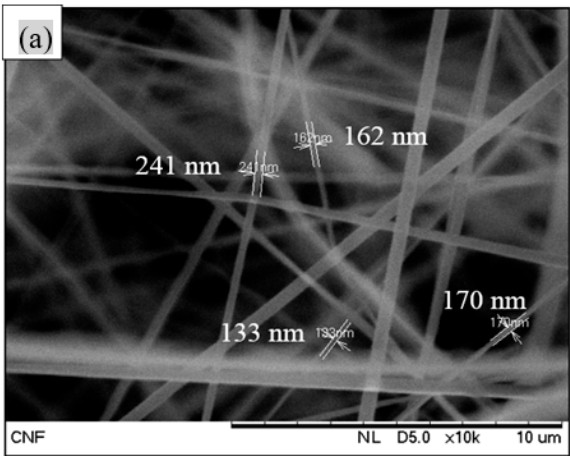
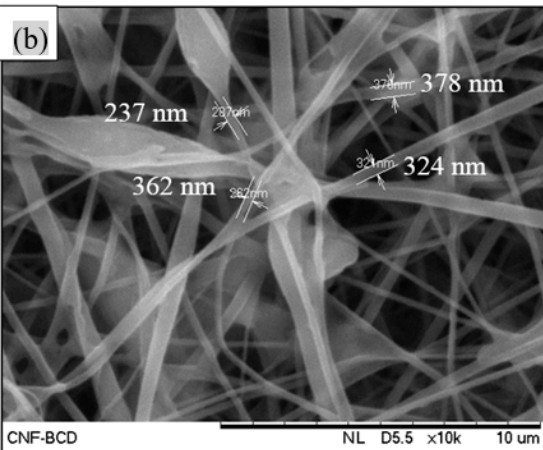

**Figure 2.** Representative images analyzed by scanning electron microscope in original magnification of 10,000 (**a**) cellulose nanofiber with average fiber diameter of 177 nm and (**b**) β-CD functionalized cellulose nanofiber with average fiber diameter of 312 nm.

### 3.4. Surface Chemical Characterization of Nanofiber

The surface chemical characterization for both types of CNF was performed by using Fourier-Transform Infrared Spectroscopy (FTIR) to further demonstrate the effect of the functionalization of CNF with β-CD as depicted in Figure 3. FTIR spectra represented in Figure 3 proves that changes occurred on the functional groups of the surface of CNF/β-CD. For the CNF, the absorption band between 3600 and 3000 cm$^{-1}$ was observed, which attributed to hydroxyl groups of cellulose [31]. Next, the peak at 1100 cm$^{-1}$ shown by the FTIR spectrum of CNF corresponds to C-C or C=C bonds. In comparison, CNF/β-CD was observed to have high intensity peak at 1740 cm$^{-1}$ which may be due to the formation of carbonyl group (C=O) stretching vibration mode of an ester bond between citric acid with CNF and β-CD. The resulting adsorption band of carbonyl group confirmed the chemical linkages between CNF and citric acid via ester bonds [32]. On the other hand,

the intensity of O-H peak at 3100–3550 cm$^{-1}$ was slightly decreased for CNF/β-CD due to the consumption of cellulose hydroxyl groups in the crosslinking reaction. The result based on this FTIR spectrum clearly indicates the effective crosslinking of citric acid with β-CD and CNF. Similar results were also reported by other studies including cyclodextrin functionalized cellulose nanofiber composites [33] as well as cellulose fiber synthesis via electrospinning and crosslinking with β-cyclodextrin [34].

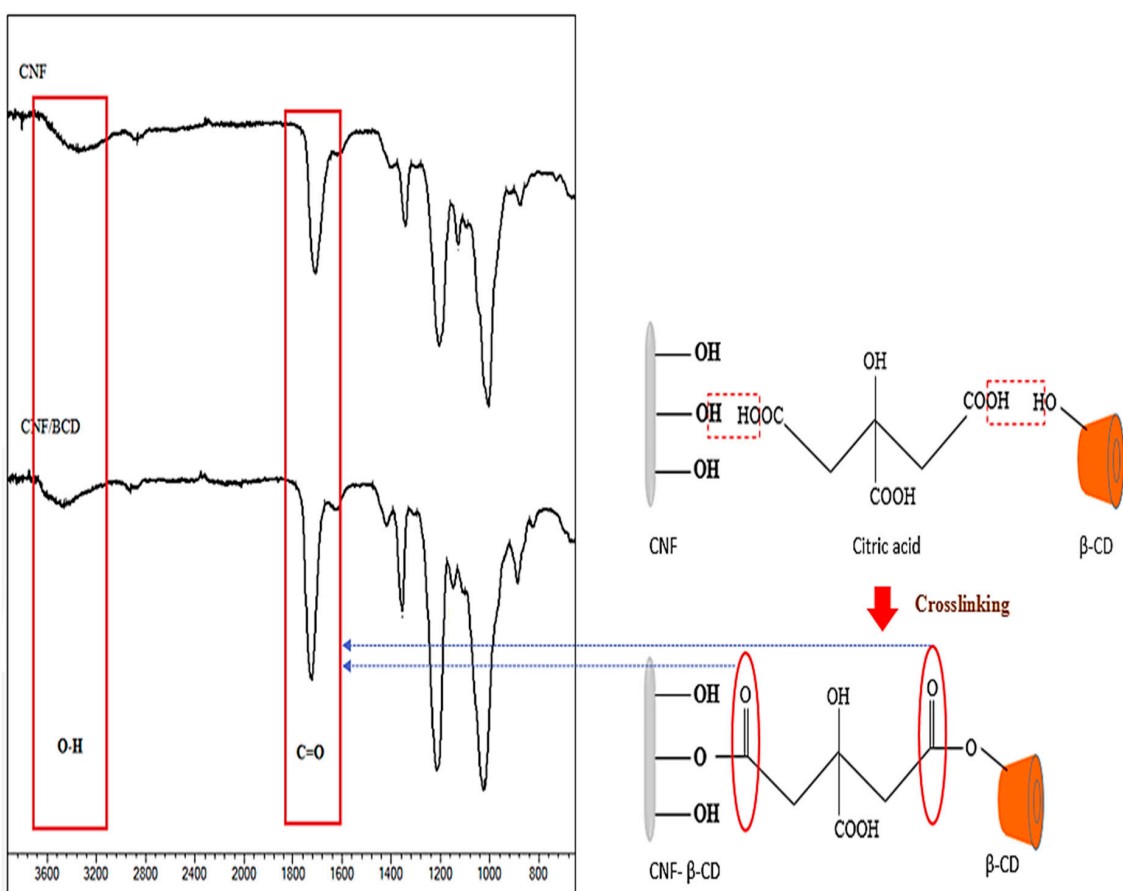

**Figure 3.** Representative Fourier-transform infrared (FTIR) spectroscopy spectra image of CNF and CNF/β-CD and the schematic diagram of the functionalization of CNF with β-CD by using citric acid as crosslinker.

### 3.5. Thermal Characterization of the Nanofibers

Thermogravimetric analyser (TGA) is used to study the thermal decomposition characteristics of the CNF and CNF/β-CD. The TGA thermograms and derivative TGA thermograms of CNF and CNF/β-CD are shown in Figure 4. It clearly shows that the main degradation of CNF occurred between 275 °C and 375 °C. For the CNF/β-CD, two major weight losses were drawn between 50–250 °C and 300–375 °C, which corresponded to main thermal degradation of β-CD and CNF. The first weight loss was attributed by the decomposition of citric acid and β-CD, which have melting points of 153 °C and 290 °C, respectively. By analysing the derivative weight percentage loss, it was recorded that the peak point of CNF at 370 °C shifted slightly to a higher temperature of 390 °C for CNF/β-CD. This slightly increased temperature of the peak points for CNF/β-CD was due to higher energy required for decomposition of the modified cellulose nanofiber that has a crosslinked structure. Increased 20 °C onset degradation temperature of CNF/β-CD compared to unmodified CNF suggested the successful modifications of the CNF. The result on increased thermal stability was also supported by cyclodextrin inclusion complexes grafted onto polyamide-6 fabric [35] as well as modification of electrospun polyester nanofibers

with cyclodextrin [27]. Modification by crosslinking methods involved either cleaving or attaching the chemical group to alter the features of the original molecule. Citric acid has been acknowledged as a good crosslinker for cellulose materials for many years and had drawn several advantages, as it is an inexpensive organic acid [36] and could react with hydroxyl groups of starch molecules through the formation of esters due to the presence of carboxylic groups in its structure [32]. The formation of crosslinks between citric acid and CNF was due to the esterification reaction. When citric acid is heated at 50 °C during functionalization process, it allows the formation the cyclic anhydride intermediate, as shown in Figure 5, that play a role as the base mechanism responsible for the development of crosslinks with β-CD and CNF. Esterification of -OH functional groups of the CNF with the cyclic anhydride intermediate lead to new carboxylic acid units, and there might also be involvement of the primary –OH groups of the polysaccharides, as they are more reactive than the secondary –OH groups in the esterification process [37,38]. The summary of comparison of CNFs characteristics is shown in Table 1.

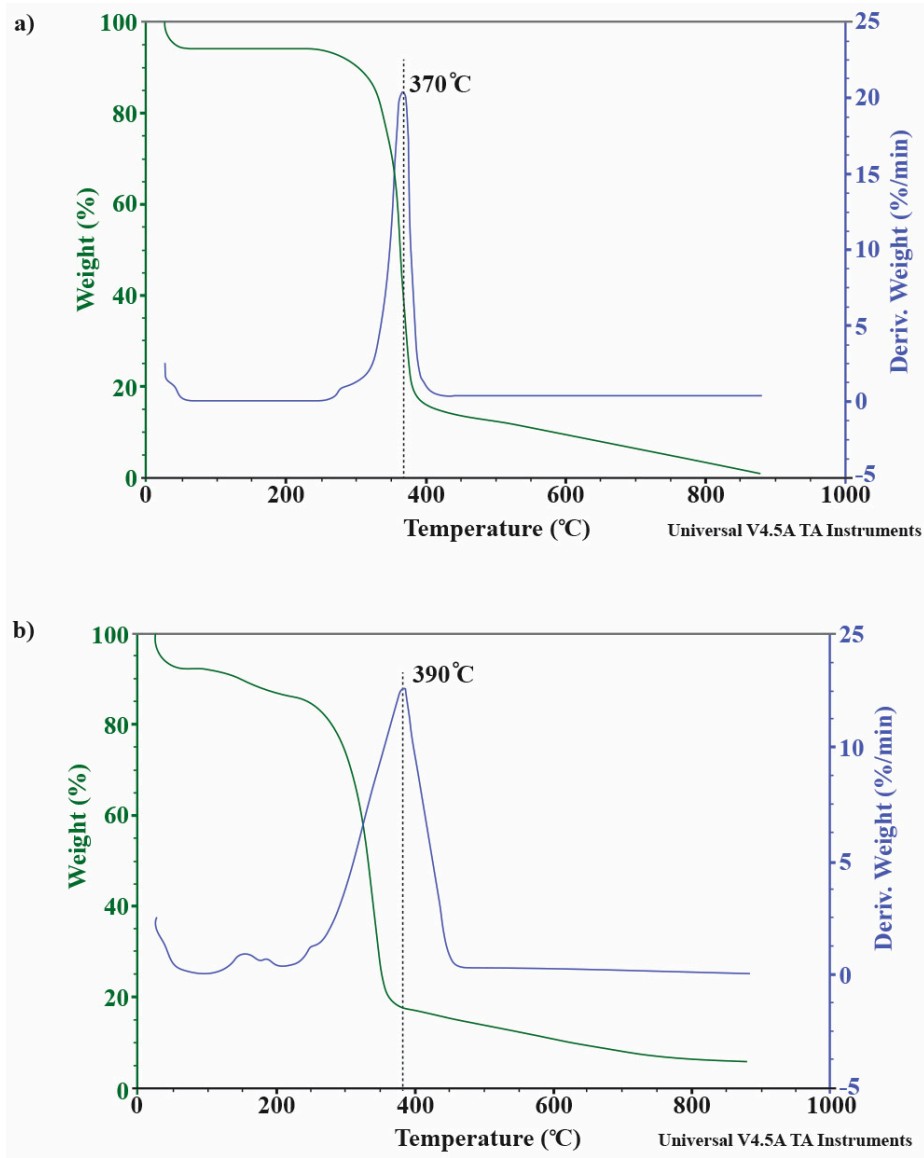

**Figure 4.** Thermogravimetric analyzer (TGA) thermograms of the (**a**) CNF and (**b**) CNF/β-CD. Green line and blue line represent the weight percentage and derivative weight percentage loss, respectively.

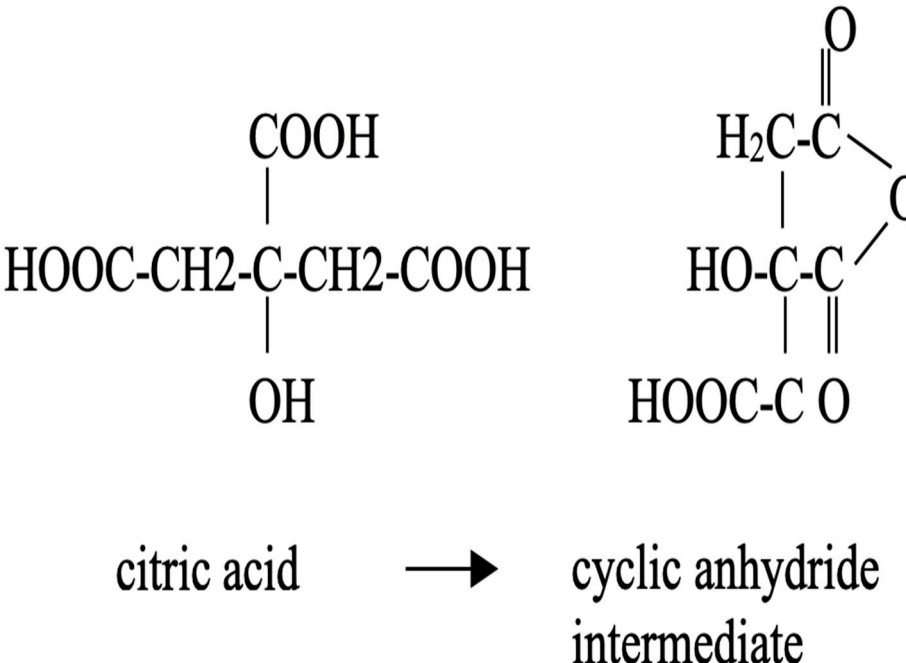

**Figure 5.** Esterification reaction of citric acid and β-CD on the CNF.

**Table 1.** Comparison of material characteristics of cellulose nanofiber (CNF) before and after functionalization with β-cyclodextrin (β-CD).

| Material Characteristics | CNF | CNF/β-CD |
|---|---|---|
| Physical appearance | Smooth surface texture | Rough surface texture |
| Morphology (SEM analysis) | Regular and uniform fiber | Irregular fiber |
| Fiber diameter (SEM analysis) | 133 nm to 241 nm | 262 nm to 378 nm. |
| Functional groups (FTIR analysis) | -OH (3600 and 3000 cm$^{-1}$)<br>-C-C or C=C (1100 cm$^{-1}$) | -OH (3100–3550 cm$^{-1}$)<br>-C=O (1740 cm$^{-1}$) |
| Thermal decomposition (TGA analysis) | 370 °C (cellulose nanofiber) | 153 °C (citric acid)<br>290 °C (β-cyclodextrin)<br>390 °C (cellulose nanofiber) |

### 3.6. *Adsorption Behavior of Palmitic Acid onto CNF/β-CD*

3.6.1. Comparison of Adsorption Performance of the CNF and CNF/β-CD

The adsorption capability of CNF and CNF/β-CD was tested using palmitic acid as a model fatty acid. Figure 6a illustrates the cumulative percentage decreases of palmitic acid concentration over time when CNF and CNF/β-CD mats have been kept in aqueous of palmitic acid at room temperature. It clearly shows that the removal of palmitic acid amount from its aqueous solution by CNF/β-CD mats was about two times better than that by CNF. The palmitic acid removal increased sharply for the first 15 min to 29.85% for CNF/β-CD and increased gradually as the time passed. However, the CNF was only able to remove approximately half the amount of those shown by the CNF/β-CD. The result obtained from the experiment indicated that the host-guest complexes between β-CD and palmitic acid was able to improve the adsorption capacity of the nanofiber by two times than those without the functionalization. The β-CD and palmitic acid complex formed almost immediately, as there were no significant differences observed on reduction of palmitic acid between 15 min and other contact times. From surface characterisation, it is noticeable that CNF have a larger average fiber diameter (AFD) and smaller surface area after modification. However, the adsorption efficiency was still further improved for modified CNF due to the β-CD structure onto nanofibers, which plays vital role in molecular capturing of palmitic

acid through host-guest interaction. When the concentration of β-CD and citric acid were compared, it showed that 7% (*w/v*) of β-CD and 8% (*w/v*) concentration of citric acid as a crosslinker were the optimum parameters needed for synthesized CNF/β-CD achieving the highest removal efficiency of palmitic acid at 15 min contact time.

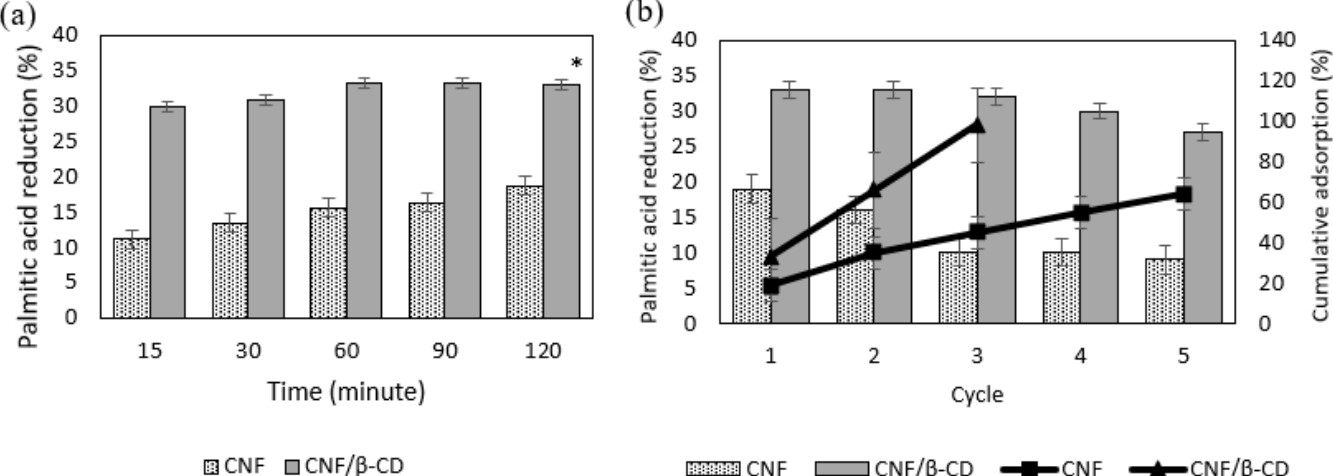

**Figure 6.** (**a**) Comparison between CNF and CNF/β-CD with 7% (*w/v*) of β-CD and 8% (*w/v*) of citric acid crosslinker on the reduction palmitic acid concentration, (**b**) Reusability analysis of cellulose nanofiber (CNF) and β-CD functionalized CNF (CNF/β-CD). The statistical significance was determined using regression data analysis and cellulose nanofibers (CNF) without functionalization as a control. $r(4) = 0.63$, * $p < 0.2$.

Reusability of CNF and CNF/β-CD were tested in a repetitive batch adsorption process. The results are illustrated in Figure 6b. Based on the presented results, significance decrease was not observed between the first and second cycle for both CNFs, as there was only a 5% drop in performance for CNF, while the performance for CNF/β-CD was maintained. Next, it was clearly seen that CNF recorded a significant drop in its performance to 47% from first cycle to third cycle. The performance of CNF/β-CD was maintained until being recycled four times, where there was significance decrease of 18% in the fifth cycle. Cumulatively, CNF/β-CD exhibited nearly 100% palmitic acid adsorption after three cycles. Meanwhile, CNF was only able to adsorb 64% of palmitic acid after five cycles. Therefore, CNF/β-CD showed remarkable adsorption capacity with very high reusability.

### 3.6.2. Adsorption Kinetics

The adsorption capacity as a function of contact time corresponding to a pseudo-first-order kinetic model and a pseudo-second-order kinetic model was studied (Figure 7a). The data were fitted into the non-linear model equations. According to the fitting results (Table 2), the $R^2$ value for the pseudo first order kinetic model is 0.6615, and this value indicated that this adsorption process does not favor this kinetic model. On the other hand, the pseudo-second-order kinetic model shows a larger $R^2$ value that is equal to 0.8983. Based on the $R^2$ values, the pseudo-second-order kinetic model is more valid and better describes time-dependent adsorption behavior for this process than the pseudo-first-order kinetic model. Thus, adsorption of palmitic acid onto CNF/β-CD possibly occurred via chemical adsorption, also known as chemisorption. The adsorption is carried out by the surface exchange reactions [39]. Palmitic acid molecules diffuse inside the CNF/β-CD where inclusion complexes, hydrogen bonds or hydrophobic interactions could take place. As a whole, this model could strongly describe the adsorption process that occurred in this case. The similar result on the pseudo-second-order kinetic behaviour was also reported before for β-CD-epichlorohydrin polymer used for removal of direct blue

(DB78) dye from wastewater, chemisorption of rhodamine B dye from aqueous solution through hydroxypropyl-β-CD cavity, as well as water-insoluble β-CD polymer for phenol uptake [40–42].

**Table 2.** Kinetics parameters for describing the adsorption of palmitic acid onto functionalized CNF at room temperature (298 K).

| Adsorption Kinetics Models | Parameters | $R^2$ |
|---|---|---|
| Pseudo-first-order | $K_1 = 0.1559\ min^{-1}$ <br> $q_e = 6940.68\ mg\ g^{-1}$ | 0.6615 |
| Pseudo-second-order | $K_2 = 6.42 \times 10^{-5}\ g\ mg^{-1}\ min^{-1}$ <br> $q_e = 7181.89\ mg\ g^{-1}$ | 0.8983 |
| Elovich | $\alpha = 1.0122 \times 10^{10}$ <br> $\beta = 0.0033\ mg\ g^{-1}\ min^{-1}$ | 0.8533 |
| Intra-particle diffusion | $K_{id} = 107.04\ mg\ g^{-1}\ min^{1/2}$ <br> $C = 5990.80$ | 0.8365 |

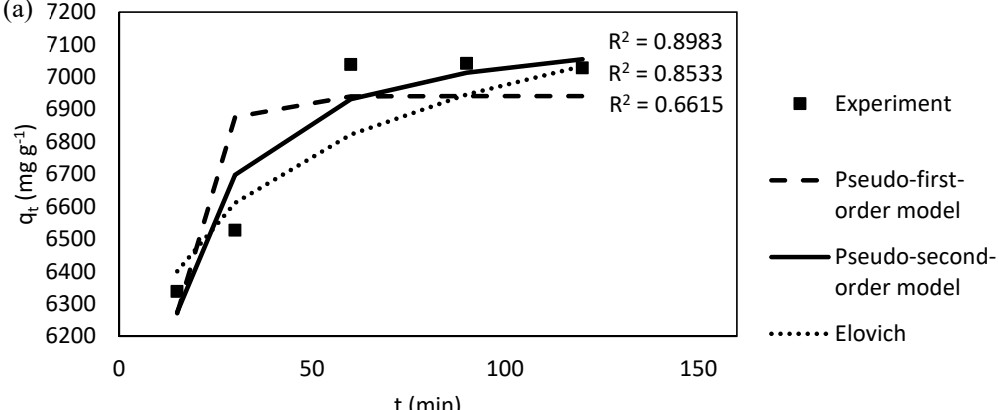

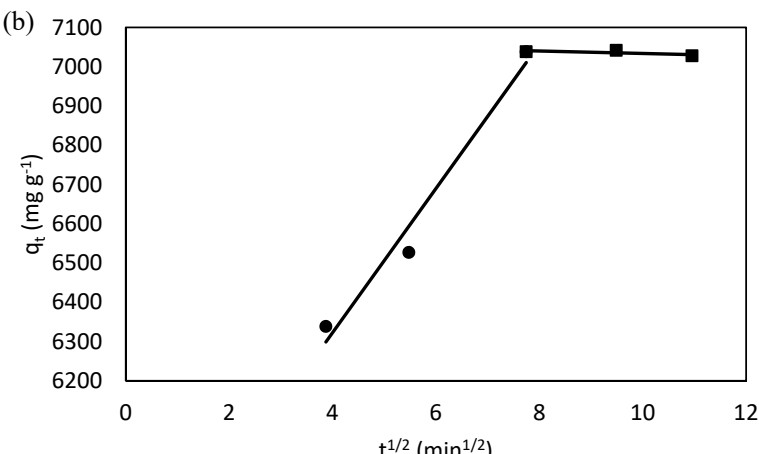

**Figure 7.** Non-linear fittings of palmitic acid adsorption onto CNF/β-CD with (**a**) pseudo-first-order model, pseudo-second-order model and Elovich model. (**b**) Linear fitting of intra-particle diffusion model.

The Elovich model (Figure 7a) also produced a good data fitting with $R^2$ value of 0.8533. Several research studies done on adsorption kinetics using the Elovich model suggested that it describes chemisorption between the adsorbent and adsorbate. For example, the adsorption between acetaminophen and chitosan/β-CD composite [43] and the

adsorption between Fe(II) and activated carbon [44]. Hence, this result further supports the chemisorption adsorption behavior as suggested by the pseudo-second-order kinetic model, with an initial adsorption rate of 0.0033 mg g$^{-1}$ min$^{-1}$. Adsorption of palmitic acid on the CNF/β-CD is chemisorption involving the hydrogen bond and host-guest interaction. This theory is strengthened by other research where the ester forms a hydrogen bond with hydroxypropyl-β-CD [45] and the removal of organic pollutants by hydroxypropyl-β-CD occurred through host-guest inclusion complexes [5].

The intra-particle diffusion model was also used to investigate the controlling step in the adsorption process. As depicted in Figure 7b, there is multilinearity indicating that there are more than one mass transfer mechanisms in the adsorption process. Furthermore, the constant C is not zero, indicating there are mass transfer mechanisms other than intraparticle diffusion. The first mechanism is chemical adsorption where the particle adsorbs from bulk solution to the β-CD active site surface of the adsorbent, forming the host-guest interaction. Fast adsorption rates can be observed in this first stage, indicating a rapid transportation of the adsorbate from the solution to the active site. The second mechanism indicates the intraparticle interaction where the mass transfer occurred on the interior sites within the pores of the adsorbent and reached equilibrium, as shown by the plateau. Pellicer et al. [46] also described the chemical and intraparticle interaction as controlling steps in the adsorption of Direct Red 83.1 onto CD-based adsorbents. Feng et al. [47] reported two main stages of liquid film diffusion and intraparticle diffusion in naproxen adsorption onto β-CD immobilized reduced graphene oxide composite.

### 3.6.3. Adsorption Isotherms

The isotherms data for palmitic acid adsorption at 25 °C were simulated with Langmuir, Freundlich, Sips and Temkin isotherm models using nonlinear fitting. The data $q_e$ against $C_e$ for experimental and simulated models are shown in Figure 8. The fitting data were tabulated in Table 3. Based on the $R^2$ value, Sips isotherm was the best to simulate the adsorption of palmitic acid onto CNF/β-CD, which proposed a monolayer adsorption process. Langmuir isotherms also gave high $R^2$ values that suggested the adsorption process underwent monolayer adsorption on homogeneous surface. The maximum adsorption capacity for palmitic acid obtained from Sips and Langmuir models were 8349.23 mg g$^{-1}$ and 10485.38 mg g$^{-1}$, respectively. In the research done on adsorption of Cu (II) on β-CD by Lv et. al. [4], the surface suggested is homogeneous as the $R^2$ value of 0.999, indicating the experimental data fitted the isotherm best. Furthermore, in the study of adsorption isotherm between organic pollutant and β-CD done by Chen et. al. [48], the result obtained is similar to this study with a high $R^2$ value of 0.999 on the Langmuir isotherm.

The separation factor, $R_L$, for the Langmuir isotherm model was greater than zero and less than one, which indicated a favourable adsorption process. Furthermore, the data fitted into the Freundlich model with the value of $n_f$ ranging between 0 to 10, which also indicates that the adsorption was favourable [46]. The value of $n_f$ (1.9444) obtained in this study was higher than one which indicates a physical adsorption [48]. The Temkin isotherm produced heat of adsorption, $b_T$ of 2.4 kJ mol$^{-1}$ (equivalent to 0.57 kcal mol$^{-1}$). Heat of adsorption value less than 1 kcal mol$^{-1}$ indicates a physical adsorption process. In contrast to the pseudo-second-order kinetic model that proposed a chemisorption process, the Freundlich and Temkin isotherm models suggest a physisorption process. The best explanation of this discrepancy probably lies on the adsorption of palmitic acid that involved the formation of palmitic acid-β-CD complex with host-guest interaction through noncovalent bonding. While most chemisorption processes were described to involve the formation of covalent bonds, the host-guest interaction between palmitic acid and β-CD molecules was associated with the formation of forces weaker than the covalent bond, such as ionic bonding, hydrogen bonding, van der Waals forces and hydrophobic interactions. Lv et al. [4] describes the plausible adsorption mechanism of bisphenol pollutants onto β-CD modified cellulose nanofiber through electrostatic interactions, π-π stacking interactions, hydrogen bonding, and hydrophobic interaction. Furthermore, the cellulose nanofiber itself

could possibly adsorb the palmitic acid through physisorption process. Further analysis is required to reveal the underlying mechanism of interaction for this study. Therefore, it could be concluded that palmitic acid adsorbs onto the CNF/β-CD via both physisorption and chemisorption.

Gibbs free energy was calculated to reveal the spontaneity of the adsorption process. The Gibbs free energy value was 1.627 kJ mol$^{-1}$ at 298 K (25°), suggesting a non-spontaneous adsorption process. The positive value of Gibbs free energy stipulated an endothermic process that requires energy input to take place. This is consistent with the positive value heat of adsorption from the Temkin isotherm model that designated an endothermic process. The adsorption process is said to be more favorable to occur at a higher temperature. The positive $b_T$ values were also reported previously in the adsorption between organic pollutants and β-CD polymers [1,48]

Generally, the experimental data fitted all the adsorption isotherms used in this study as the correlation coefficient obtained were satisfactory. The Sips model best explained the adsorption behaviour of palmitic acid onto functionalized CNF/β-CD. The Sips isotherm can describe the adsorption process the best as a monolayer process. However, multilayer interaction might as well exist as suggested by the Freundlich and Temkin isotherm models.

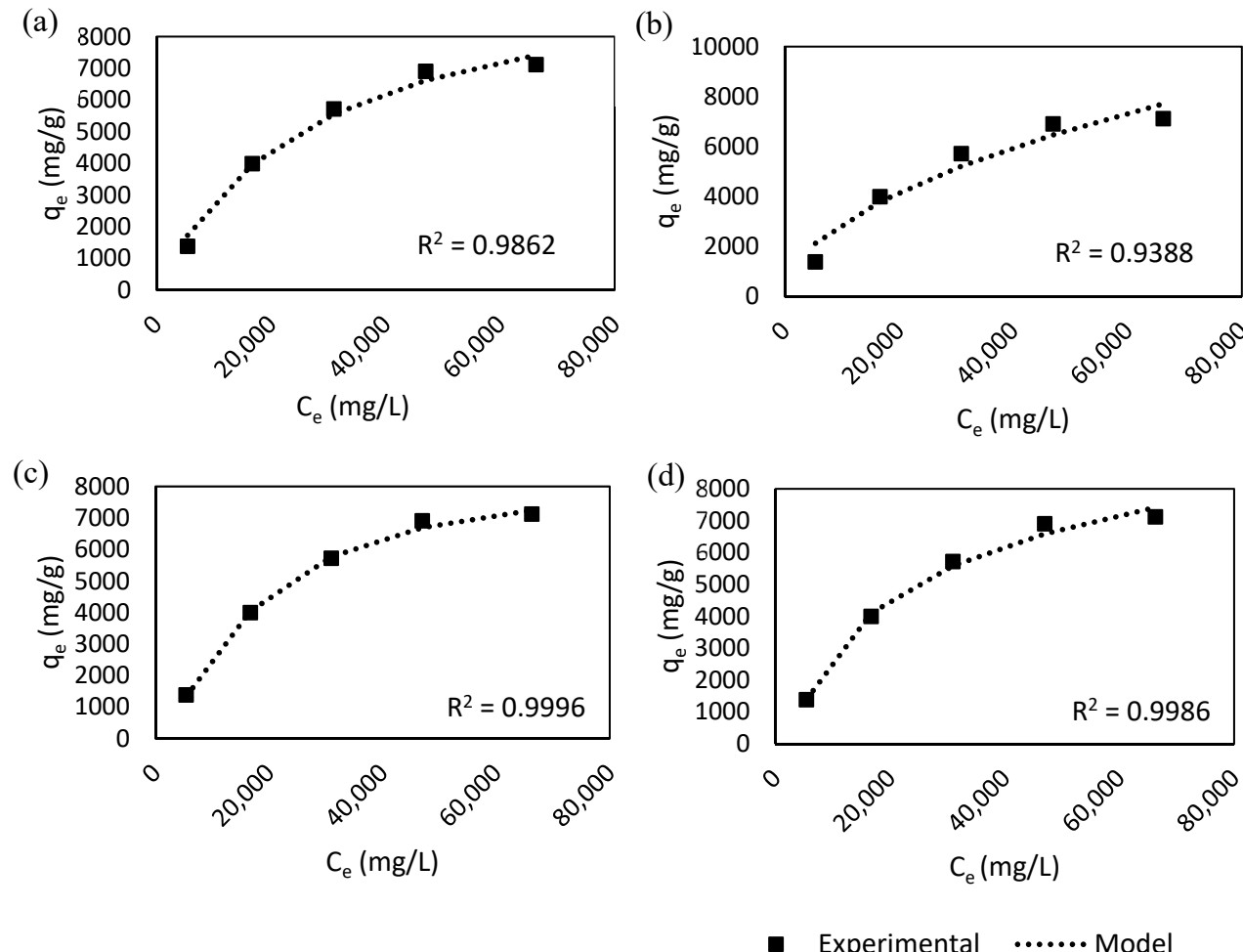

**Figure 8.** Non-linear fittings of palmitic acid adsorption onto CNF/β-CD with (**a**) Langmuir isotherm, (**b**) Freundlich isotherm, (**c**) Sips isotherm and (**d**) Temkin isotherm models.

**Table 3.** The coefficients and the constants of the Langmuir, Freundlich, Sips and Temkin isotherm models.

| Isotherm Models | Parameters | Correlation Coefficient, $R^2$ | Type of Surface/Adsorption Type |
|---|---|---|---|
| Langmuir | $K_L = 3.64 \times 10^{-5}$ L mg$^{-1}$<br>$q_m = 10485.38$ mg g$^{-1}$<br>$R_L = 0.7331$ | 0.9862 | Homogeneous/monolayer |
| Freundlich | $n_f = 1.9444$<br>$K_f = 25.5563$ mg g$^{-1}$ L mg$^{1/nf}$ | 0.9388 | Heterogenous/multi-layer |
| Sips | $K_S = 5.7141 \times 10^{-5}$ L mg$^{-1}$<br>$q_{ms} = 8349.23$ mg g$^{-1}$<br>$n_S = 1.4177$ | 0.9996 | Homogeneous or heterogenous/monolayer |
| Temkin | $K_T = 3.3 \times 10^{-4}$ L mg$^{-1}$<br>$b_T = 2402.17$ J/mol<br>(0.57 kcal/mol) | 0.9986 | Multi-layer |

## 4. Conclusions

Functionalization of β-CD has improved the properties and adsorption capacity of cellulose nanofiber, and this serves as a promising material for uptake of various pollutant molecules from wastewater, including the hydrophobic molecules. Characterization study on CNFs by SEM analysis shows the maintenance of the nanofibrous structure but with increased fiber diameter after surface modification with β-CD. The presence of β-CD layer coating on the surface of CNF was supported by FTIR analysis that revealed changes on surface functional groups after modification was made. The resulted adsorption band of the carbonyl group confirmed the chemical linkages between CNF and citric acid via ester bonds. The thermal characterization analyzed by TGA observed that CNF/β-CD has higher thermal stability, which indicated more energy is required to decompose CNF that has a crosslinked structure.

Palmitic acid removal was recorded as two times higher when CNF/β-CD was used compared to CNF. CNF/β-CD reduced its adsorption performance after five cycles, while CNF adsorption capacity reduced after two cycles. The palmitic acid adsorption onto CNF/β-CD reached equilibrium after 60 min, and the pseudo-second-order kinetic is the best model to simulate the kinetic data of palmitic acid adsorption. According to the Sips and Langmuir isotherm models, the maximum adsorption capacity was 8349.23 mg g$^{-1}$ and 10485.38 mg g$^{-1}$, respectively, at 25 °C. The isotherms indicated homogenous distribution of β-CD on the surface of CNF that leads to the uniform adsorption of palmitic acid to form a monolayer coverage.

The intraparticle diffusion model suggested chemical adsorption and intraparticle interaction as the controlling steps in the adsorption process. Thermodynamic analysis proposed the nonspontaneous endothermic process of adsorption which is favourable at higher temperatures. It can be concluded that the adsorption of palmitic acid onto CNF/β-CD was characterized as a combination of the chemisorption and physisorption processes, as the palmitic acid formed a complex with β-CD via host-guest interaction that involves noncovalent bonding. It is also speculated that the physical adsorption of palmitic acid onto CNF surfaces might also take place.

This is the first study to describe the adsorption behaviour of palmitic acid onto CNF/β-CD. This work provides an important fundamental starting point for further investigation of the adsorption mechanism of fatty acids onto CNF/β-CD and as a guideline for the practical application of oily wastewater treatment.

**Author Contributions:** Conceptualization, N.H.A.M.; Methodology, N.A.M.S. and G.Y.L.; Validation, N.I.W.A.; Formal Analysis, S.F.Z.M.F. and N.M.; Investigation, N.A.M.S. and G.Y.L.; Data Curation, M.A.H.B.; Writing—Original Draft Preparation, N.A.M.S. and G.Y.L.; Writing—Review & Editing, N.K.M.F.K. and N.H.A.M.; Visualization, N.K.M.F.K.; Supervision, R.M.I. and H.A.E.E.; Project Administration, N.H.A.M.; Funding Acquisition, N.H.A.M. All authors have read and agreed to the published version of the manuscript.

**Funding:** This study was funded by Malaysian Research University Network Young Researchers Grant Scheme, Ministry of Higher Education Malaysia (No. R.J130000.7851.4L904) and UTM Research University Grant (No. R.J130000.2651.17J89).

**Institutional Review Board Statement:** Not applicable.

**Informed Consent Statement:** Not applicable.

**Acknowledgments:** The authors would like to express their appreciation to Universiti Teknologi Malaysia and Ministry of Higher Education Malaysia for the support and sponsor involved.

**Conflicts of Interest:** The authors declare no conflict of interest.

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
