# Peer review of "Adsorption of Fatty Acid on Beta-Cyclodextrin Functionalized Cellulose Nanofiber"

_sustainability, doi:10.3390/su15021559_

Round 1

Reviewer 1 Report

(1) Normally an abstract should include the generality of the topic along with briefly stating the purpose of the study undertaken and any meaningful conclusions based on the obtained results. Hence, the abstract needs rewriting (No data is provided in the abstract).

(2) Keywords should be revised. The title words should not be repeated here.

(3) Referencing is not consistent. The whole manuscript lacks recent literature from the years 2018-2022. The literature review should be revised.

(4) The required reference for the "Synthesis of CNF by electrospinning process" section should be presented.

(5) Lines 162-179; the required reference should be presented.

(6) Only R2 is not sufficient to prove the fit quality of the isotherm and kinetic models. At least one error analysis should be used.

(7) Line 392: "Sips  model  best  explained  the  adsorption behavior of  palmitic  acid  onto" However, the qms was found to be 14285.71.  This value is not logical. The SIP model is not properly fitted to the equilibrium data. I think that the linear fitting is not suitable and should be modified.

(8) The units of the parameters in Tables 1 and 2 should be presented.

(9) The adsorption capacity is 10000 mg/g? (Table 2); please check the data

(10) A better conclusion should be provided. This section should be changed by trying to give the importance and practical implications of your results.

Author Response

Dear Reviewer,

Thank you very much for your comments and valuable suggestions to improve the quality of our work. Here are the point-to-point reply to the comments:

(1) Normally an abstract should include the generality of the topic along with briefly stating the purpose of the study undertaken and any meaningful conclusions based on the obtained results. Hence, the abstract needs rewriting (No data is provided in the abstract).

**The abstract has been revised according to the suggestion.

(2) Keywords should be revised. The title words should not be repeated here.

**The keywords have been changed.

(3) Referencing is not consistent. The whole manuscript lacks recent literature from the years 2018-2022. The literature review should be revised.

**We added some recent literature in the introduction. (e.g. Wongfaed et. al., 2020, Ekka et. al., 2022, Deaver et. al., 2020)

(4) The required reference for the "Synthesis of CNF by electrospinning process" section should be presented.

**The method was adopted from Liu & Hsieh (2002). Ultrafine Fibrous Cellulose membranes from Electrospinning of cellulose Acetate. Journal of Polymer Science. 40, 2119-2129. We added the reference no. 15.

(5) Lines 162-179; the required reference should be presented.

**All required references have been added.

(6) Only R2 is not sufficient to prove the fit quality of the isotherm and kinetic models. At least one error analysis should be used.

**We have changed the linear data fitting to non-linear data fitting as the other reviewer suggested that it could give more accurate correlations. The non-linear fittings showed improved R2 values for some models and comparable with linear fitting for the others. We included the graphs in Figure 7 and Figure 8. The new results and discussion were included in section 3.6.

 (7) Line 392: "Sips  model  best  explained  the  adsorption behavior of  palmitic  acid  onto" However, the qms was found to be 14285.71.  This value is not logical. The SIP model is not properly fitted to the equilibrium data. I think that the linear fitting is not suitable and should be modified.

**We have perform non-linear fitting for all models, and the resulted maximum adsorption capacity (qm) was 8349.23 mg g-1 and 10485.38 mg g-1 according to Sips and Langmuir isotherm model, respectively. We think these values are comparable.

(8) The units of the parameters in Tables 1 and 2 should be presented.

**The units were presented in Table 1 and Table 2 (now Table 2 and Table 3).

(9) The adsorption capacity is 10000 mg/g? (Table 2); please check the data

**The data in Table 2 (now Table 3) were changed according to the non-linear fitting results.

(10) A better conclusion should be provided. This section should be changed by trying to give the importance and practical implications of your results.

**We have revised the conclusion to include the importance and practical implications of the results.

We hope that the revised manuscript answers your comments and suggestions. Thank you again.

**author's reply

Regards,

Nor Hasmaliana Abdul Manas

Reviewer 2 Report

A study titled "Adsorption Mechanism of Fatty Acid on Beta-Cyclodextrin Functionalized Cellulose Nanofiber" reported the removal of palmitic acid from aqueous media using functionalize CNF with β-cy-clodextrin. The authors have put a lot of effort into experiments and the results shows good performance. The manuscript needs major revision before it can be published in this journal. The comments are given as follows:

Point 1: Palmitic acid was chosen as the model fatty acid, however, I did not see it in the introduction. Why was it chosen? Is it high concentration in the water? Or is it polluting? The reason why the author chose it should be clearly and fully stated.

Point 2: The experimental data is very detailed, but the layout of some figures needs to be redesigned, such as Figure 7.

Point 3: In part of adsorption kinetics and adsorption isotherms, some of the data involved in the discussion should be shown in the form of figures. In addition, the necessary parameters should also be shown in the tables and not just mentioned in the discussion section.

Point 4: The comparison of properties should be mentioned in the manuscript, and it is better to have a comparison table, which can make readers more intuitive and clear to understand the level of the material properties of the study in this field.

Point 5: Temperature is a very important parameter for adsorption properties, please add details in the experiment section.

Point 6: For adsorption materials, the ambient temperature will also affect the performance, so I suggest adding the study of adsorption thermodynamics.  

Point 7: The title of this paper contains adsorption mechanism, however, there is no clear topic of mechanism in manuscript, so I suggest the authors to prepare a new tittle based on the research work.

Author Response

Dear Reviewer,

Thank you very much for your comments and valuable suggestions to improve the quality of our work. Here are the point-to-point reply to the comments:

A study titled "Adsorption Mechanism of Fatty Acid on Beta-Cyclodextrin Functionalized Cellulose Nanofiber" reported the removal of palmitic acid from aqueous media using functionalize CNF with β-cy-clodextrin. The authors have put a lot of effort into experiments and the results shows good performance. The manuscript needs major revision before it can be published in this journal. The comments are given as follows:

Point 1: Palmitic acid was chosen as the model fatty acid, however, I did not see it in the introduction. Why was it chosen? Is it high concentration in the water? Or is it polluting? The reason why the author chose it should be clearly and fully stated.

**We have added information about palmitic acid in the introduction.

“Palmitic is one of the long chain fatty acids (LCFAs) found in high concentration in the palm oil mill effluent that is recalcitrant to biodegradation (Wongfaed et. al., 2020). It was also a major fatty acid found in industrial dairy wastewater (65%) (Ekka et. al., 2022). LCFAs at concentration higher than 0.5 mM could potentially inhibit methanogens in anaerobic digester of wastewater treatment (Sousa et al., 2013), and at concentration above 16 mM was shown to cause a lag in methane production (Deaver et. al., 2020). Therefore, there is an urgency in removing LCFAs from the wastewater.”

Point 2: The experimental data is very detailed, but the layout of some figures needs to be redesigned, such as Figure 7.

**Thank you for the suggestion. Figures were improved.

Point 3: In part of adsorption kinetics and adsorption isotherms, some of the data involved in the discussion should be shown in the form of figures. In addition, the necessary parameters should also be shown in the tables and not just mentioned in the discussion section.

**We have included the related data in Figure 7, Figure 8, Table 2 and Table 3.

 Point 4: The comparison of properties should be mentioned in the manuscript, and it is better to have a comparison table, which can make readers more intuitive and clear to understand the level of the material properties of the study in this field.

**We added Table 1 for comparison of the materials.

Point 5: Temperature is a very important parameter for adsorption properties, please add details in the experiment section.

**All adsorption experiments were conducted at room temperature (25 °C). This information has been added in section 2.5.

Point 6: For adsorption materials, the ambient temperature will also affect the performance, so I suggest adding the study of adsorption thermodynamics.  

**All adsorption experiments were conducted at room temperature (25 °C). The study of adsorption thermodynamics at other temperatures is out of this scope of study.

Point 7: The title of this paper contains adsorption mechanism, however, there is no clear topic of mechanism in manuscript, so I suggest the authors to prepare a new tittle based on the research work.

**The title is revised to remove the ‘mechanism’.

We hope that the revised manuscript answers your comments and suggestions. Thank you again.

**author's reply

Regards,

Nor Hasmaliana Abdul Manas

Reviewer 3 Report

The paper entitled Adsorption Mechanism of Fatty Acid on Beta-Cyclodextrin Functionalized Cellulose Nanofiber presents interesting results in the use of CNF functionalized with b-CD for the adsorption of palmitic acid as a model of oily wastewater adsorbent. However, many issues must be acquired for the publication of this article.

Q1 – 2.3 What about the mass of CNF immersed in the solution for functionalization? There is only sample size description, but mass is an important parameter to inquire.

Q2 – 2.5 The methodology must be extensively rewritten. Pointing out the mass of the adsorbent is essential. How were the cycle experiments performed? Were the samples isolated and washed or was the solution just changed? Was the concentration of palmitic acid the same in each cycle? Which column is used in HPLC? Several questions arise when the adsorption results are presented because there are no details of the methodology. I suggest separating item 2.5. The first would address only for kinetics and the second for the isotherms. I think it's clear this way

Q3 – I encourage authors not to use linear models for kinetic modeling and adsorption isotherms. Several articles demonstrate that linear models embed error in the data and show false correlations leading to erroneous conclusions. https://doi.org/10.1007/s11783-009-0030-7, http://dx.doi.org/10.1016/j.jtice.2017.01.024, https//doi.org/10.1016/j.cej.2009.09.013

Q4 – 3.2 The authors mention that the addition of citric acid modifies the texture of CNF and consequently the adsorption of palmitic acid but do not show any results. They could incorporate SEM images of different samples for textural comparison.

Q5 – The foundation for crosslinking between compounds is based on FTIR and TGA data. I suggest that NMR or XPS experiments data be added on behalf of two uncomfortable issues: 1) Item 3.4 FTIR justifies the structure based on the intensity differences between the O-H stretches of the CNF and the formation of new crosslink C=O. However, the simple addition of citric acid will change this ratio of intensities. The cited reference [24] uses XPS for verification and the other [25] uses XRD. 2) I don't think the derivative is correct at Figure 5b. The maximum of the derived function is far from the TGA inflection. Apparently, there is no displacement of 30 degrees after functionalization. In the text itself (line 293) there is mention of a 20-degree shift for degradation temperature. Which sample was used for this characterization? What percentage of b-CD and citric acid? Mass loss data at lower temperatures (150-250) should also be explored as may contain important information.

Q6 - Item 3.6 – Are the data used for kinetic modeling shown in figure 7? Was the time used for the isotherms 30 minutes? Make it explicit in the text

Q7 – 3.6 As there is no methodology described for reusability, there is no way to understand how results of 160% of cumulative adsorption after 5 cycles are possible. This result would be better represented as a function of acid adsorbed mass.

Q8 – 3.7 The kinetic data must be presented before the isotherms because the adsorption equilibrium time study was carried out and this was probably the time used in the adsorption experiments.

Q9 – 3.7 There is no assurance in the data in Table 1 without showing the adsorption isotherms. The modeling data show a lot of discrepancy between the obtained data. Even if the R2 values are different, the Qmax data of Langmuir and SIPs are very discrepant, the same for the n factor of SIPs and Freundlich. This demonstrates that there is no good correlation between the experimental data in this model. There is no way to know how many points were used if there is a horizontal asymptote to explore the Qmax values. I understand the use of Tenkin's model to explore the thermodynamics of the reaction, but simple experiments with temperature variation would show much more cohesive thermodynamic parameters. The authors say that the process is exothermic based only on parameter b. In lines 387 to 389 they talk about data from the literature but do not cite any.

Q10 – 3.8 Was the data n=1 of the SIPs isotherm calculated or assigned? If it was calculated, I would like it to be displayed with more decimal places. If it really is n=1, it is possible to calculate the equilibrium constant of the process by intercepting the Ceq/Qe vs Ceq graph and measuring the Gibbs energy to improve the thermodynamic discussion.

Q11 – Only two kinetic models were tested, but the authors talk about palmitic acid diffusion in CNF/BCD. Why other kinetic models were not used? If there really is diffusion, I suggest that the result of modeling with intraparticle diffusion model and Elovich be presented. The kinetic modeling graphs must also be presented because without the data there is no way to check if this is really the best model. If the pseudo-second-order model demonstrates chemical interaction, which interaction should occur? The same characterization analyzes should be performed after adsorption, mainly FTIR and, if possible, NMR.

Author Response

Dear Reviewer,

Thank you very much for your comments and valuable suggestions to improve the quality of our work. Here are the point-to-point reply to the comments:

The paper entitled Adsorption Mechanism of Fatty Acid on Beta-Cyclodextrin Functionalized Cellulose Nanofiber presents interesting results in the use of CNF functionalized with b-CD for the adsorption of palmitic acid as a model of oily wastewater adsorbent. However, many issues must be acquired for the publication of this article.

Q1 – 2.3 What about the mass of CNF immersed in the solution for functionalization? There is only sample size description, but mass is an important parameter to inquire.

**The average weight of 4 cm x 4 cm CNF mat was 0.05 g. The information was added in section 2.3.

Q2 – 2.5 The methodology must be extensively rewritten. Pointing out the mass of the adsorbent is essential. How were the cycle experiments performed? Were the samples isolated and washed or was the solution just changed? Was the concentration of palmitic acid the same in each cycle? Which column is used in HPLC? Several questions arise when the adsorption results are presented because there are no details of the methodology. I suggest separating item 2.5. The first would address only for kinetics and the second for the isotherms. I think it's clear this way

**The methodology was revised to include those important information. The mass of the adsorbent in during functionalization was 0.05 g and during the adsorption experiment was 0.1 g. These information have been added in respective methodology.

Reusability study was described in section 2.5.1. The palmitic acid concentration was the same in each cycle.

Column C18 was used in the HPLC analysis. This information was added in respective methodology.

Section 2.5 was separated to 2.5.1, 2.5.2 and 2.5.3, following the suggested sequence.

Q3 – I encourage authors not to use linear models for kinetic modeling and adsorption isotherms. Several articles demonstrate that linear models embed error in the data and show false correlations leading to erroneous conclusions. https://doi.org/10.1007/s11783-009-0030-7, http://dx.doi.org/10.1016/j.jtice.2017.01.024, https//doi.org/10.1016/j.cej.2009.09.013

**The linear data fitting was changed to non-linear data fitting. The new results and discussion were included in section 3.6.

Q4 – 3.2 The authors mention that the addition of citric acid modifies the texture of CNF and consequently the adsorption of palmitic acid but do not show any results. They could incorporate SEM images of different samples for textural comparison.

**We provided the SEM images of CNF and CNF functionalized with β-CD and citric acid as crosslinker, as comparison (Figure 2). The increase in fiber diameter and fiber irregularity for functionalized CNF were consistent with the rough surface texture feeling.

Q5 – The foundation for crosslinking between compounds is based on FTIR and TGA data. I suggest that NMR or XPS experiments data be added on behalf of two uncomfortable issues: 1) Item 3.4 FTIR justifies the structure based on the intensity differences between the O-H stretches of the CNF and the formation of new crosslink C=O. However, the simple addition of citric acid will change this ratio of intensities. The cited reference [24] uses XPS for verification and the other [25] uses XRD. 2) I don't think the derivative is correct at Figure 5b. The maximum of the derived function is far from the TGA inflection. Apparently, there is no displacement of 30 degrees after functionalization. In the text itself (line 293) there is mention of a 20-degree shift for degradation temperature. Which sample was used for this characterization? What percentage of b-CD and citric acid? Mass loss data at lower temperatures (150-250) should also be explored as may contain important information.

**Due to limited resources and facilities, the suggested analysis cannot be done and that is out of this scope of study. However, when comparing our data with references [33] and [34], we obtained similar observation on the FTIR spectra. References [33] and [34] mutually supported the FTIR results with XPS analysis for verification of the crosslink. Furthermore, the decreased intensity of O-H peak at 3100–3550 cm−1 for CNF/ β-CD observed in this study was also consistent with the cited references, that was due to the consumption of cellulose hydroxyl groups in the cross-linking reaction. If the simple addition of citric acid will increase the intensity of O-H, it does not explain the fact that O-H peak intensity was reduced in CNF/ β-CD. It was also supported by the absence of leached β-CD after rigorous washing (section 3.2). Hence, we are confident that the crosslinking process happened.

We are sorry for the typing mistake. The peak point of CNF was at 370 °C and was shifted slightly to a higher temperature of 390 °C for CNF / β-CD. The samples consisted of CNF and CNF/ β-CD with 8% (w/v) citric acid and 7% (w/v) β-CD. The weight loss at lower temperature was discussed in section 3.5.

“The first weight loss was attributed by the decomposition of citric acid and β-CD, which have melting points of 153 °C and 290 °C, respectively.”

Q6 - Item 3.6 – Are the data used for kinetic modeling shown in figure 7? Was the time used for the isotherms 30 minutes? Make it explicit in the text

**Yes, the data was used for kinetic modelling. The time used for isotherm experiment was 60 minutes. The information was added in section 2.5.

Q7 – 3.6 As there is no methodology described for reusability, there is no way to understand how results of 160% of cumulative adsorption after 5 cycles are possible. This result would be better represented as a function of acid adsorbed mass.

**Reusability method was added in section 2.5.1. The cumulative adsorption was purposely calculated to show how many cycles it needs to remove 100% of the 70000 ppm of initial concentration of palmitic acid. Since it leads to confusion, we omit the last two data that exceed 100%. The respective discussion was contained in section 3.6.

Q8 – 3.7 The kinetic data must be presented before the isotherms because the adsorption equilibrium time study was carried out and this was probably the time used in the adsorption experiments.

**The results sequence was changed according to your suggestion (section 3.6).

Q9 – 3.7 There is no assurance in the data in Table 1 without showing the adsorption isotherms. The modeling data show a lot of discrepancy between the obtained data. Even if the R2 values are different, the Qmax data of Langmuir and SIPs are very discrepant, the same for the n factor of SIPs and Freundlich. This demonstrates that there is no good correlation between the experimental data in this model. There is no way to know how many points were used if there is a horizontal asymptote to explore the Qmax values. I understand the use of Tenkin's model to explore the thermodynamics of the reaction, but simple experiments with temperature variation would show much more cohesive thermodynamic parameters. The authors say that the process is exothermic based only on parameter b. In lines 387 to 389 they talk about data from the literature but do not cite any.

**After careful checking, we found there were some calculation error in linear data fitting. The previous linear data fitting was changed to non-linear data fitting. Therefore, the improved results were presented in section 3.6. As for the temperature, we only conducted adsorption at room temperature. The variation of temperature will be conducted in the future, that is out of this scope of work.

Q10 – 3.8 Was the data n=1 of the SIPs isotherm calculated or assigned? If it was calculated, I would like it to be displayed with more decimal places. If it really is n=1, it is possible to calculate the equilibrium constant of the process by intercepting the Ceq/Qe vs Ceq graph and measuring the Gibbs energy to improve the thermodynamic discussion.

**The linear data fitting was changed to non-linear data fitting. The new results and discussion were included in section 3.6. The Gibbs free energy was calculated and discussed in section 3.6.3

Q11 – Only two kinetic models were tested, but the authors talk about palmitic acid diffusion in CNF/BCD. Why other kinetic models were not used? If there really is diffusion, I suggest that the result of modeling with intraparticle diffusion model and Elovich be presented. The kinetic modeling graphs must also be presented because without the data there is no way to check if this is really the best model. If the pseudo-second-order model demonstrates chemical interaction, which interaction should occur? The same characterization analyzes should be performed after adsorption, mainly FTIR and, if possible, NMR.

**The data was fitted into intraparticle diffusion model and Elovich. Methods were added in section 2.5 and the respective results and discussion were added in section 3.6. The kinetic graphs were presented in Figure 7 and Figure 8. The discussion on the chemical interaction involved was also added in section 3.6.3.

As for the analysis, we are sorry for not being able to commit the suggested analysis due to resources and facilities constrains.

We hope that the revised manuscript answers your comments and suggestions. Thank you again.

**author's reply

Regards,

Nor Hasmaliana Abdul Manas

Round 2

Reviewer 1 Report

This paper can be accepted in its present form.

Reviewer 2 Report

With modifications, the manuscript has been improved.

Reviewer 3 Report

The modifications requested by the reviewers improve the present work and the authors made all the requested changes. Therefore, I recommend the publication of the paper.